Corrected: Author correction

# Reflection from a free carrier front via an intraband indirect photonic transition

Mahmoud A. Gaafar [1,2], Dirk Jalas [1], Liam O'Faolain[3,4,5], Juntao Li[6], Thomas F. Krauss [7], Alexander Yu. Petrov [1,8] & Manfred Eich[1,9]

The reflection of light from moving boundaries is of interest both fundamentally and for applications in frequency conversion, but typically requires high pump power. By using a dispersion-engineered silicon photonic crystal waveguide, we are able to achieve a propagating free carrier front with only a moderate on-chip peak power of 6 W in a 6 ps-long pump pulse. We employ an intraband indirect photonic transition of a co-propagating probe, whereby the probe practically escapes from the front in the forward direction. This forward reflection has up to 35% efficiency and it is accompanied by a strong frequency upshift, which significantly exceeds that expected from the refractive index change and which is a function of group velocity, waveguide dispersion and pump power. Pump, probe and shifted probe all are around 1.5 μm wavelength which opens new possibilities for "on-chip" frequency manipulation and all-optical switching in optical telecommunications.

[1] Institute of Optical and Electronic Materials, Hamburg University of Technology, Hamburg 21073, Germany. [2] Department of Physics, Faculty of Science, Menoufia University, Menoufia 32511, Egypt. [3] SUPA, School of Physics and Astronomy, University of St. Andrews, St. Andrews, Fife KY16 9SS, UK. [4] Tyndall National Institute, Lee Maltings Complex, Dyke Parade, Cork T12 R5CP, Ireland. [5] Centre for Advanced Photonics and Process Analysis, Cork Institute of Technology, Cork T12 P928, Ireland. [6] State Key Laboratory of Optoelectronic Materials & Technology, Sun Yat-sen University, Guangzhou 510275, China. [7] Department of Physics, University of York, York YO105DD, UK. [8] ITMO University, 49 Kronverkskii Ave., 197101 St. Petersburg, Russia. [9] Institute of Materials Research, Helmholtz-Zentrum Geesthacht, Max-Planck-Strasse 1, Geesthacht D-21502, Germany. Correspondence and requests for materials should be addressed to M.A.G. (email: mahmoud.gaafar@tuhh.de) or to J.L. (email: lijt3@mail.sysu.edu.cn)

Reflection of electromagnetic waves from moving boundaries has been of interest for many years due to their potential for frequency conversion. Doppler effects occurring in the case of moving mirrors[1] and ionization fronts[2–5] have been theoretically and experimentally investigated by many authors. Nonrelativistic plasma mirrors driven by lasers have been also used to improve the temporal contrast of ultraintense pulses with reflectivities up to 70% in the fundamental wavelength of 800 nm at free carrier (FC) density of the order of $10^{21}$ cm$^{-3}$[6]. Blue shifts of several nanometres were observed in these experiments due to Doppler reflection from plasma expanding with sound velocity[7]. Ionization fronts are a more versatile route to achieving moving mirrors since lower pump power densities are necessary. In this case, only the boundary of FC plasma is moving and not the electrons themselves in vacuum. This approach is well known for frequency conversion in free space at microwave frequencies[2–5,8]. More recently, the effect has been extended to terahertz light by generating the plasma front in a silicon wafer with the FC concentration of $10^{20}$ cm$^{-3}$[9,10]. For the plasma to act as a metallic reflector, the plasma density $N$ must be high enough for the effective plasma frequency $\omega_p$ to exceed the frequency of the input wave $\omega_i$, where $N = \omega_p^2 \varepsilon_0 \varepsilon_r m / e^2$[2,9]. For a wavelength of 800 nm, for example, this corresponds to a carrier density of $1.7 \times 10^{21}$ cm$^{-3}$, while 1550 nm requires $0.46 \times 10^{21}$ cm$^{-3}$.

We now translate this exciting physics into guided wave optics and show how the waveguide dispersion can be exploited to substantially decrease required FC concentration for substantial reflection via an indirect intraband transition. Photonic crystal (PhC) waveguides, fabricated on silicon, are an appealing platform due to their compatibility with on-chip integration and their inherent flexibility in dispersion design. First, the ability to dramatically enhance nonlinear effects[11–13] by using slow light PhC waveguides[14–17] facilitates the generation of FCs by two photon absorption (TPA) using picojoule pulses. Furthermore, slow light PhC waveguides allow for tailoring the photonic bands with regions of different group velocities and dispersion[18,19]. This way, an ionization front can be generated by a pump moving with a group velocity that is different to the group velocity of the probe[20,21]. The transmission of light through the ionization front can then be described as an indirect photonic transition, whereby the wave vector and frequency of an optical probe are simultaneously altered upon a transition between two modes of a photonic structure[20,22,23]. Recently, the reflection from a counter propagating front in a slow light waveguide was theoretically proposed assuming a FC concentration of the order of $10^{19}$ cm$^{-3}$, a level that cannot be generated by TPA in silicon[24].

Here, we optimize the indirect transition for the co-propagating scheme and adjust the group velocity of the pump that generates FC plasma. We manage to achieve the case when light stays in the same band of the unaltered waveguide and thus undergoes an indirect intraband transition. As a result, we demonstrate a forward reflectivity of 35% from the plasma front. The plasma front is generated by the TPA of a 6 ps-long pump pulse with an on-chip peak power of 6.2 W and an induced FC density of $7 \times 10^{17}$ cm$^{-3}$. We investigate the reflectivity via the interaction of a continuous wave (CW) probe wave co-propagating with the plasma front inside a 400 μm-long slow light waveguide. Only a portion of the CW probe is interacting with the front and experiences indirect transition. To explain the transition, we split the CW probe into wave packets of finite duration and follow their trajectories in time and space. Some of the wave packets initially propagating slower than the plasma front, are bounced and accelerated, finally escaping from the front in forward direction. The forward reflection of the probe wave packets (i.e. the "bouncing") is accompanied by a frequency upshift. In this respect, the effect is similar to the optical analogue of event horizons induced by Kerr nonlinearity[25–27]. We also conduct our experiments using a CW probe light, as it simplifies the experimental verification of the frequency shift and reflection efficiency. A narrow CW spectrum facilitates spectral separation between the probe and shifted probe wavelengths. The portion of CW light interacting with the front can be determined from the group velocities of the pump and probe and the length of the slow light waveguide. This way the energy at the shifted frequency can be directly compared to the energy in this portion of CW light resulting in the conversion efficiency. The experimental results are confirmed by numerical simulations.

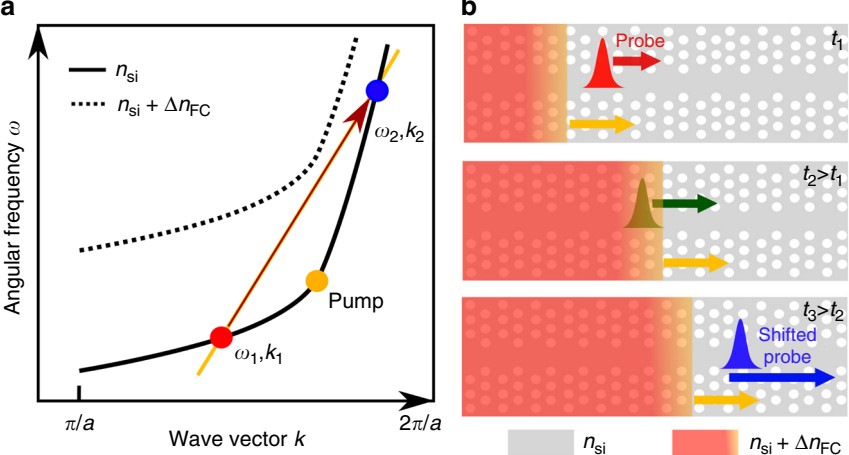

**Fig. 1** Schematic of the interaction between front and probe. **a** Schematic representation of an intraband indirect photonic transition in a slow light silicon PhC waveguide. The solid curve represents the dispersion band of waveguide mode in the ground state (with refractive index $n_{si}$), while the dashed curve indicates the switched state with refractive index $n_{si} + \Delta n_{FC}$. We use the PhC band with positive group velocity to represent the dispersion relationship. The orange line represents the phase continuity line with a slope equal to the group velocity of the pump pulse set at the knee of the dispersion band (orange dot). The red and blue dots indicate the initial (slow mode) and final (fast mode) states of a probe wave, respectively. **b** Schematic of the experiment. A pump pulse generates FCs in the silicon by TPA and, consequently, induces a change of refractive index which propagates with the velocity of the pump pulse. The region with the red colour gradient corresponds to the rising edge of the front. The orange arrow indicates the velocity of the pump pulse, while red, green and blue arrows indicate the velocities of a wave packet of the probe at different times, respectively

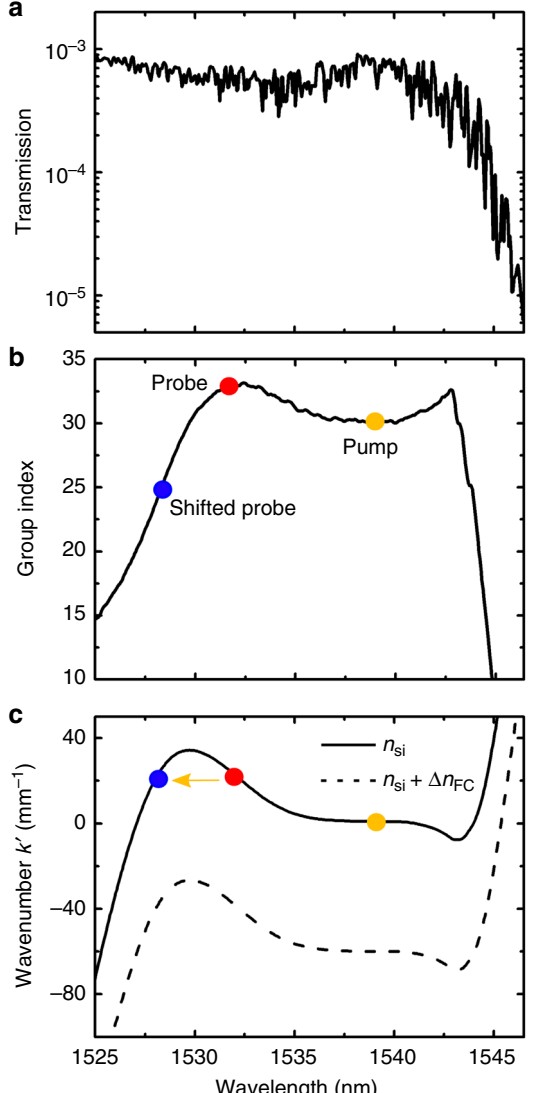

**Fig. 2** Characteristics of the fabricated slow light silicon PhC waveguide. **a** Linear transmission and **b** measured group index. **c** The calculated dispersion band from the measured group index in **b**. The wave vector corrected by the slope of the pump group velocity is presented. Due to very small band diagram shift the curves cannot be presented in the original form similar to Fig. 1a. Solid curve represents the original dispersion band of waveguide mode with refractive index $n_{si}$, while the dashed curve indicates the waveguide mode with refractive index change of $\approx -3 \times 10^{-3}$. Dots indicate the locations of the input wavelengths of probe wave (red dot), pump pulse (orange dot) and the expected output wavelength of the shifted probe wave (blue dot) after the intraband transition took place. The orange line represents the phase continuity line with a slope equal to the travelling velocity of the pump pulse. Due to the slope correction, this phase continuity line now appears horizontal

We also show that the FC plasma front is asymmetric in its reflection properties. It will accelerate and bounce a slow probe wave packet propagating in front of it, yet it will transmit a fast probe wave packet approaching from behind.

## Results

**Intraband transition.** After injection of the pump pulse into the PhC waveguide, the pulse modifies the optical properties of the waveguide by generating FCs via TPA. In the switched zone, the refractive index of silicon, $n_{si}$, reduces by a quantity $\Delta n_{FC}$ that

is proportional to the FC density $N_{FC}$, which in turn changes and blue shifts the dispersion curve. Therefore, a refractive index front constitutes a moving boundary between two PhC zones with slightly different band diagrams[20]. When a probe wave packet is launched into the waveguide, the changes of its wave vector and frequency that are induced by the interaction with the co-propagating front are determined by the dispersion curve of the system, the propagation velocity of the front and the initial position of the probe wave vector and frequency in the band[20].

Here, we are interested in the particular situation where the probe wave packet ahead of the front cannot find states on the band of the switched PhC behind the front. Thus, the state of the probe wave, after interacting with the moving front, must remain in the initial band, which means that an intraband transition takes place. This intraband transition manifests itself as a forward reflection from the front. The basic concept to induce this transition is schematically shown in Fig. 1a. The phase continuity line defines possible states, which satisfy a continuous phase at the refractive index front. Thus, the moving front can only excite states that lie on the phase continuity line. The ratio of the possible frequency shift $\Delta\omega$ and the wave vector shift $\Delta k$ is equal to the front velocity. This relation can be derived using the phase conservation under Lorentz transformation[20], phase evolution integrals[24] or from the Doppler equation[25]. Subsequently, by utilizing the flexibility in dispersion designs, a variety of indirect transitions can be envisaged by changing the intersection points of phase continuity line with the bands of the PhC before and after the front. We are interested in the particular situation when the phase continuity line does not cut through the band of the perturbed PhC and thus there is no states behind the front that can be excited.

The intraband transition can be achieved by setting the pump pulses at the knee of the solid band at the frequency where group velocity corresponds to the slope of the phase continuity line, as illustrated in Fig. 1. The absence of states in the perturbed waveguide force the probe wave to stay in the unswitched zone of the PhC and is thus reflected at the moving front. This reflection is accompanied by a large frequency shift which, most importantly, can be achieved with a small band shift. The final state of the probe $(\omega_2, k_2)$ is determined graphically from the crossing point of the phase continuity line and the solid band. Figure 1b shows a schematic illustration of this process. The fascinating fact is that a probe wave packet initially propagating slower than the approaching pump pulse, upon interaction with the moving front, which is dragged by the faster pump pulse, is accelerated and finally escapes from the moving front in the forward direction.

However, implementing the configuration shown in Fig. 1a by choosing the pump pulse to lie at the knee of the dispersion band has some experimental drawbacks. First, due to the high intensity of the pump pulse and its centre frequency close to that of the probe wave, it is difficult to detect the shifted probe after interaction. Second, it is also challenging to distinguish the intraband transition from third-order nonlinear processes, such as four wave mixing (FWM), which would cause spectrally similar signals. Thus, the pump should be positioned at some other frequency where the group velocity is the same as the slope of the phase continuity line.

**Dispersion-engineered slow light waveguides.** Normally, slow light waveguides are optimized to obtain large bandwidth with constant low group velocity[18,19]. However, slow light PhC waveguides can also be engineered to obtain a dispersion relation with equal group velocities at three different frequencies[19]. This engineered waveguide can be used to excite pump pulses with the

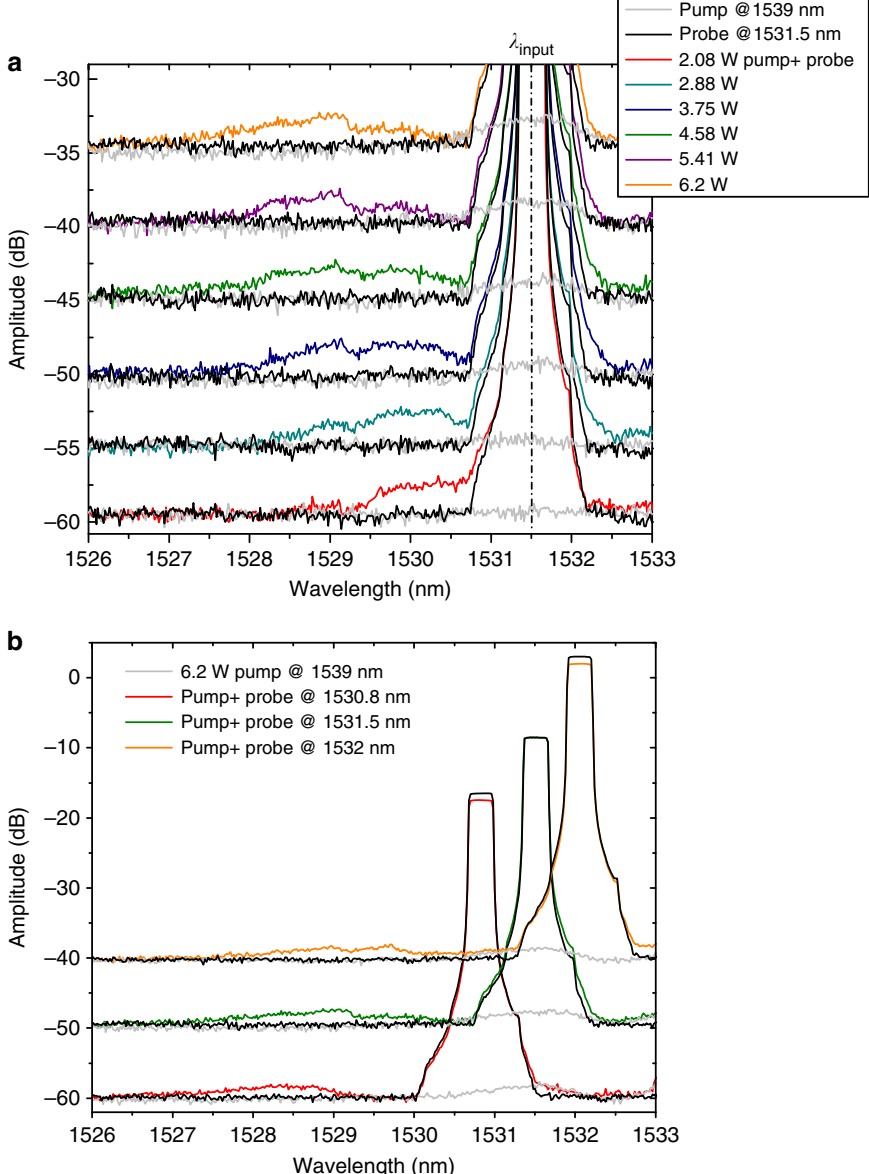

**Fig. 3** Experimental spectra recorded at the output of our 396 μm-long slow light silicon PhC waveguide. **a** Pump pulse peak power-dependent output spectra for the blue-shifted light of an input probe wave at a wavelength of 1531.5 nm. Shown is the spectrum of the CW probe that has not interacted with the pump pulses (black traces), the spectrum of the pump pulses alone (grey traces), and shown are the spectra of both the pump pulses and the CW probe (other colours). **b** Spectra for different input probe wavelengths. The different curves in **a**, **b** are shifted by 5 and 10 dB, respectively for clarity, the noise level corresponds to −60 dBm

required group velocity at a frequency distant from the initial and final frequencies of the probe. A similar dispersion relation was used in refs. [21,28] using a chirped PhC waveguide; however, due to both stronger dispersion and short length (200 μm) of the waveguide it is too challenging to induce an intraband transition in such a structure. It should be noted that both publications are focused on different effect to that discussed here and intraband transitions and reflection from the front were not investigated. The chirp in the structure further complicates the interpretation of the results obtained in refs.[21,28].

Thus to implement intraband transitions, we fabricated a single line defect PhC waveguide consisting of a hexagonal lattice of air holes in silicon. The silicon PhC waveguide has a length of 396 μm and is connected to the edges of the chip by polymer waveguides which are converted by inverse tapers into 3 μm wide silicon access waveguides. The detailed design and measurement

method of these waveguides were given in refs. [19,20], (see also Supplementary Note 1). The measured linear transmission (including coupling loss) and the group index of the TE-mode of the waveguide are shown in Fig. 2a, b, respectively. Orange and red points in Fig. 2b indicate the wavelengths and group indices of pump pulse and probe wave, respectively. As clearly seen from the spectrum, the pump pulse at ≈1539 nm has a velocity matched wavelength at ≈1529.5 nm $\left(n_g^f = 30\right)$. If we set the probe wave to 1531.5 nm, which is a bit longer than this matched wavelength, it will propagate with a slightly smaller group velocity than the refractive index front moving with the pump pulse. We will then expect an intraband transition to 1528.5 nm (Fig. 2, blue dot), which is the shortest of the three interacting wavelengths. The shifted probe wave at 1528.5 nm employs the smallest group index and therefore is going to escape from the approaching refractive index front, while staying in the unswitched zone, thus

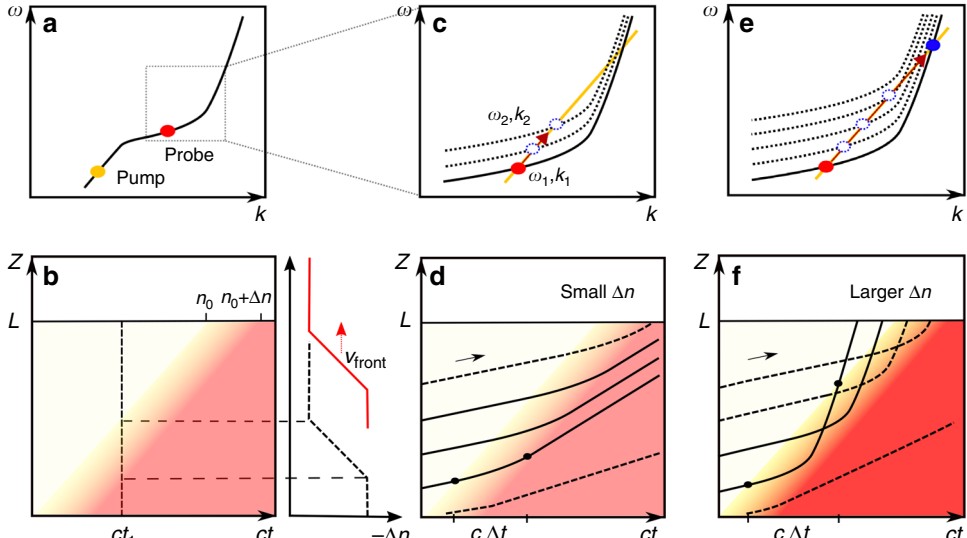

**Fig. 4** Schematic representations of the induced interband and intraband photonic transitions. **a** Schematic representation of the dispersion band of our PhC waveguide. Red and orange dots indicate the locations of the input probe wave packets and the pump pulse, respectively. **b** Spatiotemporal change of $\Delta n$. The red filling colour indicates the z-coordinate range of the waveguide where the refractive index has been switched already (switched zone). The graded reddish area indicates the fact that the time function of the pump pulse power has a finite steepness, thus the refractive index front is gradual in any position as the pump pulse transforms the waveguide medium. The two horizontal dotted lines represent the rising length of the index front. The velocity of the refractive index front, indicated in **b**, **d**, **f** as the slope of any line of equal reddish colour tone in the graded part, is the same as the group velocity of the pump pulse thus equal to the slope of the band at pump frequency. **c**, **e** show the dispersion bands and the corresponding interband and intraband photonic transitions of the input probe caused by insufficient (dashed blue dots) and by sufficient (solid blue dots) values of $\Delta n$, respectively. **d**, **f** Schematic illustrations of the probe wave signal trajectories for the induced transitions in **c**, **e**, respectively. Black lines in **d**, **f** represent wave packets of the CW input probe. Solid lines illustrate the wave packets which undergo complete interband **d** and intraband **f** transitions, while the dashed lines depict the wave packets which only undergo incomplete transitions. The black dots in **d**, **f** mark the points where the wave packet enters and exits the front

in the unswitched band. This arrangement is different to the scenario which features the pump pulse wavelength or frequency in between the respective values for the probe and shifted probe waves—easy to confuse with a FWM process. The modified scenario is now based on the pump pulse wavelength being the longest one and the shifted probe wavelength being the shortest one. This choice of parameters clearly rules out a FWM process which, otherwise, would have competed with our novel intraband transition. Figure 2c shows the calculated dispersion band corrected by the slope corresponding to the pump pulse group velocity. Original curves are presented in the Supplementary Fig. 3. Due to very small band diagram shift and small group velocity mismatch the curves are difficult to consider in the original form. Thus, the slope corrected wavenumber $k'$ at a frequency $\omega$ is calculated as $k'(\omega) = k(\omega) - k(\omega_{\text{pump}}) - (\omega - \omega_{\text{pump}})/v_{\text{f}}$. The orange line represents the phase continuity line with a slope equal to the travelling velocity of the pump pulse[20], which leads to a horizontal line in the corrected representation. By choosing the group index of the probe wave to be $n_{\text{g}}^{\text{s}} = 33$, we expect an intraband transition with a blue shift of $\approx 3.1$ nm. The intraband transition discussed here can be obtained for the aforementioned group velocity of the pump only. In addition, the group velocities of both pump and probe waves should be close to each other.

**Experimental observation of intraband transition**. A six picosecond long pump pulses derived from 100 MHz repetition rate mode locked laser are launched into a 396 µm-long slow light silicon PhC waveguide at a centre wavelength of 1539 nm with a group index of $n_{\text{g}}^{\text{f}} = 30$. We also feed in the low power probe as a CW of light, which co-propagates with the index front in the waveguide with a slightly slower group velocity.

Figure 3 shows the experimental output spectra recorded for the transmitted CW probe wave power of $\approx 6$ µW with and without the pump pulses. Black traces correspond to the output spectra of the CW probe that has not interacted with the pump pulses, grey traces relate to the pump pulses alone, and other colours refer to the cases when both the pump pulses and the CW probe wave were present (the curves are shifted for clarity, the noise level is $-60$ dBm). Figure 3a demonstrates the measured output spectra for the blue-shifted probe wave at a wavelength of 1531.5 nm (group index of $n_{\text{g}}^{\text{s}} = 33$) as a function of the pump pulse on-chip peak power. With the pump pulses present, spectral components clearly appear which are blue-shifted with respect to the initial probe wavelength. Wave packets of the CW probe were approached by the index front and were transmitted through it or reflected from it in forward direction.

The group velocity of the pump pulses is fixed, and, accordingly, the slope of the phase continuity line is defined as well. Therefore, the induced transition of the probe wave is governed by its initial wavelength location on the band diagram and by the magnitude of the band shift. We estimate that the pump power at the input of the slow light waveguide is reduced by coupling losses of approximately $-4$ dB[20]. When the pump pulse peak power at the input of the access waveguide is low ($\approx 2$ W), i.e. when $\Delta n$ is small, we can only notice one blue-shifted peak at $\approx 1530$ nm, which is attributed to the interband transitions into the slightly shifted band into which the phase continuity line cuts (Fig. 4c). However, when we increase the pump pulse power, the band shift also increases and we can observe the appearance of another peak with a much larger blue shift. At sufficient $\Delta n$, the phase continuity line can reach the shorter wavelength position in the same band without cutting into the shifted band (Fig. 4e). This intraband transition is realized at the estimated pump pulse peak power at the input of the access waveguide of

6.2 W and pulse duration of 6 ps, which leads theoretically to FC generation of $7 \times 10^{17}\,\text{cm}^{-3}$ and a refractive index change of approximately $-3 \times 10^{-3}$ at the input of the slow light waveguide (see Supplementary Note 2).

The spectral response changes from interband regime to intraband regime not abruptly. As we discuss later, due to the pump decay and finite length of the waveguide some wave packets experience interband transitions and some intraband transitions. With increasing power, number of those probe wave packets subject to intraband transition increases, while the number of those wave packets that underwent the interband transitions decreases. As the pump pulse power further increases, the wavelength shift saturates as additional pump power only further separates the bands but has no effect on the starting and end points in the band diagram, which indicate the intraband transition. The center wavelength of the maximum blue-shifted peak is evaluated to be at $\approx$1528.5 nm. We calculate the centre wavelength of the shifted probe spectrum using Eq. (3) in ref. [20]. The measured wavelength shift of $-3$ nm matches the value of $-3.1$ nm we expect from the calculated dispersion band in Fig. 2c. In the time domain, a probe wave packet which is travelling slower than the index front is converted to a wave packet travelling at a new frequency. The shifted probe then travels faster in the unswitched zone of the waveguide than the approaching front (Fig. 2b, c) and thus escapes in forward direction. The intraband transition presented here is not reversible. A probe wave launched at 1528.5 nm from behind the pump pulse travels faster than the pump pulse, undergoes an interband transition and turns out to be blue shifted again as it penetrates the refractive index front and dives into the unswitched zone. These results are presented in the Supplementary Note 4.

Figure 3b displays the recorded output spectra of the blue-shifted probe wave for three different input probe wavelengths at 1530.8, 1531.5 and 1532 nm. The group velocities of these input probe wave packets are always slower than the group velocity of the pump pulse, thus, than the velocity of the index front. As expected from the high steepness of the band shown in Fig. 2c, the maximal wavelength of the shifted probe wave is not significantly affected as the input probe was slightly tuned towards longer wavelengths. A maximum blue shift of $\approx$3.4 nm is achieved for the input probe at wavelength of 1532 nm. This is a more than three times larger wavelength shift than that observed with indirect interband transitions[20].

## Discussion

Here we discuss the results obtained in Fig. 3 and give an explanation for the observed blue-shifted probe wave signal. Figure 4a shows a schematic representation of the dispersion band of the PhC waveguide. Red and orange dots again indicate the locations of the input probe wave and the pump pulses, respectively. The spatiotemporal change in $\Delta n$ is schematically demonstrated in Fig. 4b (left). The red filling colour indicates the $z$-coordinate range of the waveguide where the refractive index has been switched already (switched zone). As the time evolves to the right the switched zone dimension extends to positive $z$-coordinate values. Eventually, the waveguide is completely switched. As the FC lifetime in silicon PhC waveguides is in the order of 100 ps[28,29], we neglect recombination in the time frame of the front propagation. The graded reddish area indicates the fact that the time function of the FC concentration employs a finite steepness. The two horizontal dotted lines represent the spatial width of the index front. The gradual profile of the front can be explained as follows. The FC generation rate via TPA is proportional to local pump power squared (see Supplementary Note 2). The generation rate defines the slope of the FC concentration in time, and, taking the group velocity of the pump into account, in space. Thus, the maximal slope corresponds to the position of the peak power of the pump and the front duration corresponds to the duration of the pump power squared distribution. Disregarding the FC decay, the FC concentration function is the integral of the pump power squared function. In this consideration we also neglect Kerr effect as for 6 ps pulses the FC dispersion is the dominant mechanism of the refractive index change[30].

The velocity of the refractive index front, indicated in Fig. 4b, d, f as the slope of any line of equal reddish colour tone in the graded part, is the same as the group velocity of the pump pulse and thus is equal to the slope of the band at pump frequency and slope of the phase continuity line. The induced transitions of the input probe wave into the photonic bands (Fig. 4c, e) and the corresponding trajectories of its wave packets (Fig. 4d, f) for two different maximal $\Delta n$ values are illustrated, respectively.

To explain the obtained results, we choose to view the CW probe wave as being composed of individual wave packets located at different $z$-coordinate positions at a given point in time. These wave packets then interact with the front differently, depending on their relative positions to the front at a given time. Their trajectories are shown in Fig. 4d, f by solid and dotted black lines. Before the faster approaching index front encounters the wave packet, this wave packet moves along a straight line with constant group velocity of the probe. Upon contact with the front, the probe wave packet first penetrates into the zone of reduced refractive index and thus experiences an indirect transition to the new frequency and wave vector. This leads to change of the group velocity which determines how the wave packet further propagates. If the maximal refractive index change of the front is sufficient to achieve an intraband transition to a band diagram position of larger group velocity, then the probe wave packet penetrates the front up to a position where its group velocity has gradually increased to match that of the front. After that, the probe wave packet starts to recede from the front, again, as its group velocity further increases and it escapes in the forward direction. A useful approximation of the wave packet trajectory inside the front can be derived for the case where the band of the unswitched waveguide upon switching is just vertically shifted in frequency proportionally to the refractive index change and, assuming further, that the front has a graded refractive index, linearly growing along the negative $z$-direction. In this case, it can be shown that the wave packet trajectory in time and space has the same shape as the band in wave vector and frequency coordinates. This correspondence comes from the equivalency of the slopes in both curves as $\mathrm{d}\omega_{\text{band}}/\mathrm{d}k_{\text{band}}$, and $\mathrm{d}x/\mathrm{d}t$ are both the group velocity of the wave packet. Also, the frequency change of the wave packet along the trajectory is directly proportional to the time spent inside the front $\Delta t$, i.e. the duration between the points in time when the wave packets enter the front and the points in time when they leave it (e.g. black dots in Fig. 4d, f)[31]:

$$\Delta\omega = v_{\text{f}}\Delta k = \Delta\omega_{\text{shift}} \times \Delta t/\tau, \tag{1}$$

where $v_{\text{f}}$ is the front velocity, i.e. the group velocity of the pump pulse. $\Delta k$ is wave vector change, $\Delta\omega_{\text{shift}}$ is the maximal shift of the band diagram produced by the front and $\tau$ is the rising time of the front. In our experiments, the pump pulse duration and thus the rising time of the front is fixed, but the maximal band shift increases as the pump pulse power increases.

First, we consider the situation when $\Delta\omega_{\text{shift}}$ is small, i.e. at low pump pulse power (red curve in Fig. 3a). In this case, the initial band only shifts slightly to a higher frequency and a probe wave packet with an initial state $(\omega_1, k_1)$, upon interaction with front, moves along the phase continuity line to a final state $(\omega_2, k_2)$ in

the shifted band (Fig. 4c). This is the interband transition case. The expected wavelength change is relatively small and the corresponding group velocity change inside the front as well. The group velocity change in this case is insufficient to accelerate wave packets away from the front. Thus, the wave packets are transmitted through the front into the switched zone where they further propagate in the forward direction behind the front, at a slightly higher frequency and at a slightly higher group velocity as compared to before. After the injection of the front, some of these probe wave packets will undergo complete transitions (solid lines) as they experience maximal possible time in the front. However, there are other probe wave packets which undergo incomplete transitions (dotted lines). The incomplete transitions are attributed to the fact that the packets are either already inside the refractive index front at the input (lower dotted line), or they exit the slow light waveguide before complete transition could take place (upper dotted line).

Next, the situation where the intraband transition occurs and the probe wave packets escape from the front is considered. The band diagram shown in Fig. 4e indicates that the intraband transition can be realized easily when $\Delta\omega_{\mathrm{shift}}$ is large (the band shift is large). This behaviour suitably explains the occurrence of the intraband transition in Fig. 3a as we increase the power of the pump pulses. The trajectories of the probe wave packets for this case are illustrated in Fig. 4f.

In the presented description we did not take into account the decay of the pump power in the slow light waveguide due to linear or nonlinear absorption and due to scattering. Such losses diminish the FC concentration thus lead to the reduction of $\Delta\omega_{\mathrm{shift}}$ along the waveguide. We estimate total losses of $\approx 4\,\mathrm{dB}$ inside our slow light waveguide as a result of $\approx 2\,\mathrm{dB}$ linear losses[32], and $\approx 2\,\mathrm{dB}$ nonlinear losses due to FC absorption and TPA[12]. As the FC generation rate via TPA is proportional to the local pump power squared, then the index front has 8 dB exponential decay during propagation inside the waveguide. Such losses reduce the total effect of intraband transitions along the full waveguide length. The losses of the pump in combination with the incomplete transitions, lead to the spectral broadening of the blue-shifted probe shown in Fig. 3.

Now, the result in Fig. 3b can also be discussed in more detail. We mentioned before that due to the high steepness of the band shown in Fig. 2c, the maximal wavelength of the shifted probe wave should be slightly blue shifted as the input probe is slightly tuned towards longer wavelengths. However, what we achieved is a slight red shift of the shifted probe. This happens as the length of the waveguide or the time the signal spends inside the front is not sufficient to induce the complete transition. We can estimate from Eq. (1) the maximum possible wavelength shift of the probe in the structure, by assuming that the maximal time spent inside the front $\Delta t$ is equal to the travelling time of the probe inside the 400 μm structure, which is 40 ps. The index front and thus the front slope are decaying, while propagating in the waveguide. In this case the maximum possible shift of the signal can be calculated using $\Delta\omega = (\Delta\omega_{\mathrm{shift}}/\tau)\int_0^{40\,\mathrm{ps}} e^{-t/t_0} \times \mathrm{d}t$, where $1/t_0$ is a decay constant corresponding to $-8\,\mathrm{dB}$ in 40 ps. With a refractive index change of $\approx 3 \times 10^{-3}$ at the input and a rising time of the front of 6 ps, the maximum possible shift will be $\approx -3.5\,\mathrm{nm}$, which fits to the maximal obtained experimental shift. We can see from the exponential decay in the integral that longer waveguides will not help much in increasing the maximum possible shift.

We also observe a peak generated only by the pump pulses at approximately 1531.5 nm. This peak can be attributed to the special case of the dispersion wave (DW)[26,27,33,34]. We explain the appearance of this DW by self-induced indirect transition of wave packets from the pump pulses.

As we use a CW probe, only a small fraction of the total probe light can be converted within the finite length of PhC waveguide. The fraction $\eta$ of the CW probe wave power that enters the front and is frequency converted is:

$$\eta = \nu_{\mathrm{rep}}\left[\left(\frac{n_{\mathrm{g}}^{\mathrm{s}} - n_{\mathrm{g}}^{\mathrm{f}}}{c}\right)L + \tau\right], \qquad (2)$$

where $\nu_{\mathrm{rep}}$ is the repetition rate of the pulses, $c$ is the velocity of light in vacuum, $L$ is the length of the PhC waveguide and $\tau$ is the rising time of the front. The first part in the bracket corresponds to the part of CW probe that entered the front within the waveguide length $L$ and the second term relates to the CW probe wave that entered the waveguide when the front has just partly evolved for a fraction of its rise time $\tau$. We estimate the rise time of the front as $\tau = 6\,\mathrm{ps}$ approximately the duration of the pump pulse. For $\nu_{\mathrm{rep}} = 100\,\mathrm{MHz}$, $n_{\mathrm{g}}^{\mathrm{f}} = 30$, $n_{\mathrm{g}}^{\mathrm{s}} = 33$ and $L = 396\,\mathrm{\mu m}$, the maximally transformable fraction of the incoming CW probe power due to the interaction with single pump pulses arriving at 100 MHz repetition rate is $1 \times 10^{-3}$ and this value corresponds to a 100% conversion efficiency. From the measurements shown in Fig. 3(a) a pulsed pump with peak power at the input of the access waveguide of 6.2 W acts on the CW probe with transmitted power of $\approx 6\,\mathrm{\mu W}$ and causes a forward reflected and frequency transformed signal pulses with average power of $\approx 2$ nW, which is calculated by integrating over the shifted signal below the wavelength of $\approx 1529.5\,\mathrm{nm}$ and above $-60\,\mathrm{dBm}$ level. This transformed power corresponds to a fraction of the transmitted CW probe power of $\approx 3.5 \times 10^{-4}$. The ratio of this transformed probe power fraction to the maximally transformable probe power fraction of $1 \times 10^{-3}$ equals 35% and marks the actual efficiency of the intraband transition due to reflection at the propagating plasma fronts. While the amount of transformed probe power caused by interband transitions with final velocity slower than the front is estimated to be approximately 5%. The missing 60% could be attributed to the absorption of the probe wave packets by FCs during their interaction with the plasma front, as is shown in the simulations below.

Next, we simulate the FC front and CW probe wave packets interaction inside the 400 μm long waveguide with parameters corresponding to experimental conditions. We launch wave packets and follow their trajectories in a ray tracing approach. The details of the simulations are presented in the Supplementary Note 6. No fitting parameters were used. Figure 5 presents the simulation results for $\Delta\omega_{\mathrm{shift}}$ of 0.12 THz (0.9 nm) at the input corresponding to refractive index change of $\approx -3 \times 10^{-3}$. Figure 5a shows the calculated average power of the wave packets in frequency intervals of 1 nm scanned over the spectrum, the same resolution that was used in the experiment with an optical spectrum analyzer. We can see that most of the wave packets energies are concentrated in the frequency interval corresponding to intraband transitions with group velocities larger than the front velocity. The trajectories of the probe wave packets with respect to the front velocity, which are represented by black lines, before and after the interaction with the index front inside the 400 μm-long waveguide, are shown in Fig. 5b. Due to the small group velocity mismatch between the initial probe wave packets and the index front, the trajectories are difficult to be shown clearly in their original form. Thus, the slope corrected curves are presented with time $t'$ calculated as $t' = t - x/v_{\mathrm{f}}$. In this case, the two orange lines represent the index front with a rise time of 6 ps, appearing static in this representation, whereas the shading represents the front refractive index changes exponentially decaying during propagation inside the waveguide. For

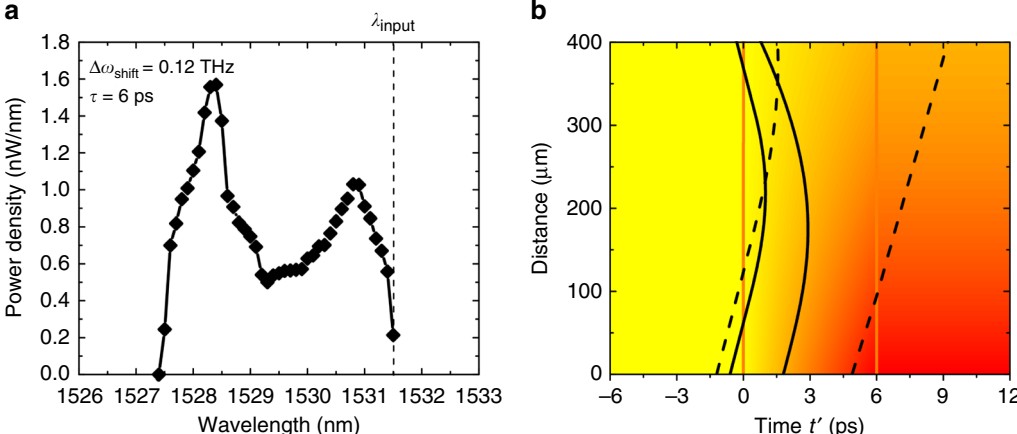

**Fig. 5** Simulation of the FC front interaction with CW probe wave packets via ray tracing. **a** Wave packets average power spectrum for 6 μW average power of the CW probe at 1531.5 nm, rising time of the front of 6 ps, and initial band shift $\Delta\omega_{shift}$ of 0.12 THz. Other parameters are listed in the Supplementary Note 6. **b** The trajectories of the probe wave packets with respect to the front velocity. The two vertical orange lines represent the extent of the index front with a rise time of 6 ps, while the shading represents the refractive index front as it exponentially decays during propagation inside the waveguide. Solid black lines represent those wave packets that undergo complete transitions, while the dashed lines depict the wave packets which only undergo incomplete transitions

comparison, original uncorrected curves are presented in the Supplementary Fig. 4.

Here we show the trajectories of only four wave packets for clarity. Solid black lines represent those wave packets that undergo complete transitions, while the dashed lines depict the wave packets which only undergo incomplete transitions. We can see clearly that after interaction with the front some wave packets are accelerated by the front and bounced in the forward direction (solid lines) and some other wave packets still have velocities smaller than the front (dashed lines). In addition, we can observe that some wave packets starting initially inside the front are accelerated but do not have the enough time to leave the front.

For the spectral evaluation we have launched 120 equally spaced wave packets with starting time in the interval $t' = [-5\,\text{ps}, 7\,\text{ps}]$. From Fig. 5a, we obtain $\approx 2.7\,\text{nW}$ average power of the shifted light, which corresponds to 45% conversion efficiency, compared to 40% conversion efficiency obtained from the experiment (35% intraband and 5% interband transitions). In the simulation we took into account the FC concentration dependent absorption of the probe. The absorption coefficient is 30 cm$^{-1}$ for FC concentration of $7 \times 10^{17}$ cm$^{-3}$, corresponding to refractive index change of $\approx -3 \times 10^{-3}$[35], and for a slow-down factor of 10 present in the waveguides considered[12]. In case of a sharper front, the time spent by the wave packets inside the front $\Delta t$ will be shorter and the wave packets will experience less FC absorption. We show by simulation (see Supplementary Note 6, Supplementary Fig. 5) that using 1 ps pump pulse with the same peak power at the input of the slow light waveguide two times higher efficiency for the intraband transition could be obtained than in case of 6 ps pump pulse. The forward reflected signal does not penetrate the region with high FC concentration behind the pump pulse, thus FC absorption does not pose a fundamental limit on the conversion efficiency of the intraband transition.

It should be also mentioned that since a CW probe was used here to determine the reflection efficiency, only 0.1% of CW wave was converted. All power of the CW probe at times between the pump pulses adds to the average detected power, thus lowering the "apparent" conversion efficiency while the "effective" conversion efficiency of the probe wave packets participating in the conversion process remains at 35% for 6 ps pump pulses. For probe pulses or pulse trains that have durations equal to the

interaction time with FC front we expect the conversion efficiency to approach 100%.

We would like to discuss here briefly the advantages of the frequency shift induced by the FC front in comparison to idler generation in the FWM configuration. First, the time span of the probe wave shifted to new frequency is not limited by pulse duration as in FWM but depends on both the group velocity differences between the pump and the probe and on the interaction length. Thus, with a FC front we can envisage to shift large portions of probe, for example, a packet of binary optical signal information. For example, if we assume a 1 mm-long slow light waveguide and a group index difference between the pump and the probe of 30, then the time span of the converted probe will be 100 ps. For a wavelength-division-multiplexing (WDM) system 0.8 nm wavelength shift would be sufficient to switch between two channels. In this case smaller pump power can be used as in this experiment and thus TPA induced pump depletion can be avoided. This value is much larger than what can normally be achieved in a FWM process, which is limited by the picosecond, or shorter, pump pulse durations. Recently, FWM has been demonstrated in a silicon PhC waveguides using a CW light for both pump and probe waves; however, the conversion efficiency is in the order of $\approx -30$ dB[36]. In addition, for switching in WDM systems, the pump can be positioned outside of the signal frequency window, such that pump has no frequency components at the signal frequency. In comparison, the FWM approach requires the pump frequency always to lie between signal frequency and shifted frequency thus additional measures would be required to filter the pump out of the signal window and spectral cross-talk between the high power pump and the signal is difficult to avoid. The third advantage is the fact that to change the frequency shift neither power nor frequency of the pump has to be adjusted. The different frequency shifts can be achieved with the same pump in waveguides with different dispersion. These advantages make the intraband optical transitions a viable option for all optical signals routing in WDM networks[37].

In conclusion, we have experimentally demonstrated a 35% optical-wave reflection and $-3.4$ nm wavelength shift by interaction with a FC plasma front generated, confined and propagating inside a 400 μm-long dispersion engineered silicon slow light PhC waveguide. The front was generated by TPA of 6 ps-long pump pulse with a peak power of 6.2 W. The forward

reflection of the probe wave packets co-propagating with the plasma front are accompanied by a frequency upshift. The presented reflection becomes possible due to a novel indirect intraband optical transition in a slow light waveguide. Under these special conditions the incident light does not find states beyond the front and has to reflect from it. The effect presented here is based on the fascinating dispersion properties of carefully engineered silicon slow light PhC waveguides which allow to control the speed of the plasma front with respect to the group velocity of the probe wave packets. We show also that the FC plasma front is asymmetric in its reflection properties. It will accelerate and reflect a slow probe wave packets propagating in front of it and will transmit a fast probe wave packets initially behind it. Due to the fact that the pump, probe and shifted probe are all at 1.5 μm wavelength, the presented effect opens new possibilities for frequency manipulation and all optical switching in optical telecommunication.

## Methods

**Experiments**. The optical measurement proceeds by launching 6 ps-long pump pulses derived from 100 MHz repetition rate mode locked fibre laser into a 396 μm-long slow light silicon PhC waveguide at a centre wavelength of 1539 nm with a group index of $n_g^f = 30$. We also feed in the low power probe as a CW of light, which co-propagates with the index front in the waveguide with a slightly slower group velocity (energy velocity). More details about the experimental setup can be found in the Supplementary material and in ref. [38]. The occurrence of an intraband indirect transition in our waveguide can be verified by recording the optical spectra of the output probe light with and without pump pulses present and as function of the pump pulse power.

**Simulation of CW wave packets trajectories**. In our simulations, we focus on the part of the incident probe light that interacts with the front and ignore both the unperturbed input CW probe light, as well as the components that are slightly down shifted in frequency due to slow FC decay after the front propagation. We assume that we can split the CW probe into wave packets and track their trajectory and frequency shift. At the end of the waveguide we calculate the energy of the shifted probe by adding up the energy of the shifted wave packets. This approach neglects the spectral width of the wave packet and interference effects between the wave packets and thus represents a ray optics approximation.

In our simulation we use a front with a linear slope. We took into account the FC absorption of the probe scaling linearly with FC concentration. The absorption coefficient is 30 cm$^{-1}$ at FC concentration of $7 \times 10^{17}$ cm$^{-3}$[35], taking a slow-down factor of 10 into account. Also, we consider an exponential decay of the front refractive index change during propagation inside the waveguide due to scattering and TPA and FC absorption with decay constant corresponding to 8 dB decay in 400 μm waveguide. We start with a defined frequency of the wave packet and calculate its wavenumber and group velocity at the input in accordance to its position in respect to the front. In a small time step a wave packet propagates a small distance according to its group velocity. Being inside the front the wave packet accumulates a frequency shift according to Eq. (1), a wavenumber shift according to $\Delta k = \Delta \omega / v_f$ and is subject to absorption according to the local FC concentration. This leads to a change of the group velocity that determines how the wave packet further propagates. Thus, we track the signal wave packets in space and time and accumulate the frequency shift. At the end at the output of the waveguide we obtain the wave packets with their new frequency and smaller energy due to FC absorption.

**Data availability**. The data that support the findings of this study are available from the corresponding author upon request.

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

## Acknowledgements

M.A.G, D.J., A.Y.P. and M.E. acknowledge the support of the German Research Foundation under grant no. 261759120, and appreciate the support of CST, Darmstadt, Germany, with their Microwave Studio Software. M.A.G, D.J., A.Y.P. and M.E. acknowledge the support of Michel Castellanos Muñoz in preparing the grant proposal. J.L. acknowledges the supports of the Ministry of Science and Technology of China (2016YFA0301300) and National Natural Science Foundation of China (11761131001, 11674402). LOF acknowledges support form the Science Foundation Ireland under Grant SFI12/RC/2276.

## Author contributions

M.A.G. performed the simulations and conducted the experiments, J.L. and L.F. fabricated the PhC waveguide, T.F.K., A.Y.P. and M.E. supervised the project, M.A.G, A.Y.P., D.J., T.F.K. and M.E. analysed the results and wrote the paper. M.A.G is the corresponding author of this paper, and J.L. is the corresponding author regarding slow light sample fabrication.

## Additional information

**Competing interests:** The authors declare no competing interests.

