## [Peer Review File · Nature Communications]

Reviewers' comments:

Reviewer #1 (Remarks to the Author):

In this paper, the original waveguide-based method is proposed to generate relativistically moving plasma mirrors with lower pump powers and lesser carrier concentration. Without vacuum conditions, the efficient reflection of 35% is obtained experimentally for a carrier concentration of $5 \times 10^{17}/\text{cm}^3$. The paper is technically sound and suitable to be published in Nature Communications with some revisions as suggested below:

- [1] In the manuscript, the reflection is up to 35%. Compared to other reports, it will be low relatively. The factors related to the reflection should be discussed, and the reasons for the reflection only up to 35% may be given.
- [2] In Fig. 2(b), the group index obtained in experiment is provided. Is it consistent with the theoretical calculation of ?
- [3] (Line 349) The author should further demonstrate that why the front has a graded refractive index, linearly growing along the negative z-direction.
- [4] The author should simply discuss potential application of optical switching.
- [5] The author should discuss influence of the decay of pump power on functions of the slow-light waveguide.
- [6] The author should clarify why the cross-talk problem can be overcome in the present design.

Reviewer #2 (Remarks to the Author):

The authors report on frequency conversion at telecom wavelengths in slow-light photonic crystal waveguides, employing free carriers produced by two photon absorption of a ps pump pulse. The mechanism for frequency conversion is the interaction of a probe pulse with the plasma generated by the pump, which shifts the waveguide dispersion resulting in a photonic intraband transition. The authors quote a 35% forward reflection efficiency at a peak power of 6.2 W, equivalent to $1.2 \times 10^9 \text{ W}/\text{cm}^2$ incident on the waveguide. The 35% efficiency is the ratio between the actual conversion efficiency, namely 3.5×10^{-4} , and the maximally transformed probe power fraction of 1×10^{-3} for the given experimental conditions. The authors claim the mechanism of intraband transitions to be a promising possibility for all-optical signal routing in WDM networks.

While this manuscript reports on advanced and well performed experiments, with interesting results for frequency conversion in slow-light PC waveguides, I do not believe this work to be important enough for publication in a Nature journal. My specific comments and objections are detailed below:

1. The title is "Reflection from a relativistic free carrier plasma mirror...", the abstract and the introduction start with a description of free carrier plasmas used as tunable mirrors at solid surfaces. Yet this work refers to a totally different situation, namely frequency conversion through interaction between a probe and a pulse beam, both propagating in a photonic crystal waveguide. It is clear that the threshold for frequency conversion is lowered by orders of magnitude compared to the surface geometry, as the beams are laterally confined and propagate for a nearly mm length across the waveguide. Moreover, "reflection" sounds as a misnomer in the present configuration, as the frequency-shifted pulse is propagating in the forward direction and is not "reflected". I find the abstract and the introduction to be misleading in this respect.
2. The results for frequency conversion by intraband transitions in a slow-light PC waveguide are interesting, but not as groundbreaking as the introduction would suggest. Indeed, this work builds on other papers on the same topic, both by the same and other authors (e.g. Refs. 27, 28, 31).

Specifically, the configuration for frequency conversion is similar to that of Ref. 27. The results for the conversion efficiency are indeed good compared to other systems, but in the end the absolute conversion efficiency is lower than 10^{-3} and most of the probe intensity is lost by absorption. It is difficult to envisage how this mechanism could be employed for all-optical routing.

3. I do not really understand why the authors present their results by writing that "we have experimentally demonstrated a 35% optical wave reflection and -3.4 nm wavelength shift by interaction with a relativistic FC plasma mirror..." The optical wave is not reflected, and the conversion efficiency is 3.4×10^{-3} .

Other remarks:

4. What happens in the results if the pump power is increased beyond 6.2 W? Can the conversion efficiency be further increased, or do absorption losses prevail?

5. Figure 4 is very hard to read, black and blue colors with such thin lines are difficult to distinguish.

Reviewer #3 (Remarks to the Author):

The paper by Gaafar et al. proposes an experimental demonstration on a photonic crystal waveguide platform of a pulse routing effect analogous to a reflection from a "free carrier plasma mirror". Its main claim, notably with respect to the previous work by Baba's team (Kondo 2014 PRL) on Dynamic Wavelength Conversion, is to achieve intraband transitions rather than interband ones, owing to the large band shift and the specific elbow-shaped bands. The free-carrier origin is, in my understanding, the reason why the term "reflection" is used, even though the pulse is forward kicked rather than backward kicked by the moving carrier plasma.

The paper is not fully convincing, and has a few weaknesses however, some more severe than the others, with respect to its claims. In my opinion, it cannot be published as such unless the main weaknesses are properly and sufficiently addressed.

From the point of view of wording first, the paper title and some introductory paragraphs boast the "relativistic" aspect. It certainly adds appeal to the narrative and admittedly the situation bears a fair relationship, but it should be much weighted with the peculiarities of this experiment : the velocity is only $v \sim c/40$, much smaller than c . Furthermore, a moving front of nonlinear matter modification due to somehow relaxed electrons in the carrier plasma has been commonplace in nonlinear optics, be it in fiber, in crystals, or in nanophotonic devices. This contrasts, for instance, with what the recent Ref. 7 states for electrons of relativistic mirrors: "When pulled outwards, they form relativistic electron jets (red arrow in Fig. 1a) that are responsible for the ROM[Relativistic Oscillating Mirror] attosecond pulse emission." It is not clear whether that part of the relativistic physics is present in the reported work. So, much more care would be needed and possibly the title could be modified if the case is not made more strongly towards the analogy. Ref.28 by Kondo and Baba does not invoke relativistic effects at all, while the mechanisms could be with little reduction claimed to be "half the same" (interband vs. intraband, say) insofar as plasma effects are concerned.

The fact that the experiment works in transmission rather than in reflection also causes some difficulty in pursuing the analogy. In balancing the discussion more clearly towards the "ionization fronts", the paper could gain more conviction than within the present unbalanced choice of emphasizing relativistic plasma mirrors.

However, the two main weaknesses of the paper are in my opinion the following ones :

- First, the paper (including the SI) is based on a single sample and a single length. Practitioners often make several samples side by side, it is hard to argue that the task of multiple sample fabrication was an obstacle. Given the limits on spectral behaviour entailed by the particular length choice (and this is according to the authors themselves: conversion is partial due to finite length etc., the role of dispersion should be interesting with different dispersions, line 466, etc.), the limit of a single sample experiment leaves more room for some alternative explanations (e.g. based on trapped/untrapped photons, possible role of irregularities on the waveguide causing some unsteady ionization front dynamics, etc.) than if an experiment with a different sample could take place. To some extent, changing the wavelength gives extra room, but the wavelength ranges where the expected phenomena unfold are very limited given the specific mechanism targeted and the associated dispersion demands. If a longer device of identical dispersion, for instance, would raise difficulties such as too low transmission, then there are some limits to the proposed scheme with state-of-the-art PhC waveguides, and they should be clearly and quantitatively acknowledged if not emphasized.

- Secondly, the methodology is rather weak when it comes to analysing the resulting probe spectra, mainly those of Fig.3. Experimentally, was there no way to on/off modulate the probe beam (at 100 MHz) to avoid, say, 90% of the background ? More importantly, an attempt to analyse the spectral shape should be explicitly made and rendered as e.g. graphs/bars with one point associated to each spectrum/pump power, so that other investigators in the future can check what is going on based on the sole non quantitative spectral shape appearance. Indicating the pulse "centers" (vaguely defined) by arrows in such a smooth spectral context is good at a preliminary stage (for a post-deadline paper, for instance) but more explicit analysis is commendable for a high-standard paper.

- A few other suggestions to attain an easier reading are the following ones:

1) it would be good to still attempt (in the SI for instance) to represent the band structure at rest and with carrier plasma, even if it is graphically awkward (local zooms at elbows?). The present effort to grasp the diagrams with renormalized slope are truly difficult even for a scientist familiar with PhC band structures.

2) Δ_n is negative, so Fig.4(b) is also awkward to read as the usual trend in optics is that $\Delta_n > 0$ slows down light, while here $\Delta_n < 0$ speeds it up. Check and possibly amend the "index axis" of Fig.4b to clarify the signe and role of Δ_n .

3) Fig.4c, the solid blue dot is still interband, the caption is probably misleading (or the color ...).

4) Typos : line 332 "we chose to view *THE* cw probe".

Some other choices are idiosyncratic (to my eyes) and not the best:

--The use of the verb "employ" to describe the plasma rising front "finite steepness".

--The use of "a.k.a." on line 358.

--The sentence lines 385-386 : "Then, ... such is ..." looks awkward.

--Lines 463 464 : "thus caus*ING* a bigger cross-talk problem. [And] *The* third advantage ... " (suppress "And").

Overall, my recommendation, again, is not to publish as such, essentially because the paper too often indulges in easier lines of analysis for critical aspects: the relevance of the relativistic mirror comparison, the degree of conviction of the conclusion drawn on a single sample, and the rather qualitative analysis of the spectra (even if no new science is involved in fitting waveforms among themselves or against some specific profiles).

But it must be recognized that a distinctive intraband transition was indeed observed which makes a possibly large difference with respect to earlier work, so that a revision with substantially

enhanced analysis with a more appropriate relationship to current trendy topics in optics could be envisioned.

Reviewers' comments:

Reviewer #1 (Remarks to the Author):

In this paper, the original waveguide-based method is proposed to generate relativistically moving plasma mirrors with lower pump powers and lesser carrier concentration. Without vacuum conditions, the efficient reflection of 35% is obtained experimentally for a carrier concentration of $5 \times 10^{17}/\text{cm}^3$. The paper is technically sound and suitable to be published in Nature Communications with some revisions as suggested below:

[1] In the manuscript, the reflection is up to 35%. Compared to other reports, it will be low relatively. The factors related to the reflection should be discussed, and the reasons for the reflection only up to 35% may be given.

Authors: We thank the reviewer for his comment. The missing amount of transformed probe power is attributed to the free carrier absorption of the probe wave packets during the interaction with the front. We have added results on wave packet simulations in the main text and supplementary materials taking into account free carrier absorption (pages 20-21 in the main text and section 5 in supplemental materials). Our numerical results reproduce the experimental results without fitting parameters.

The reflection efficiency can be increased if the time spent by the wave packets inside the front Δt is shorter and hence the wave packets will experience less free carrier absorption. This can be achieved by using a sharper front. We showed by simulation that using 1 ps pump pulse with the same peak power at the input of the waveguide will lead to a reflection efficiency of 70%. We added to the following text to the manuscript in page 21:

“In case of a sharper front, the time spent by the wave packets inside the front Δt will be shorter and the wave packets will experience less FC absorption. We show by simulation [see supplemental materials, Fig. S5], that using 1 ps pump pulse with the same peak power at the input of the slow light waveguide two times higher efficiency for the intraband transition could be obtained than in case of 6 ps pump pulse. The forward reflected signal does not penetrate the region with high FC concentration behind the pump pulse, thus FC absorption does not pose a fundamental limit on the conversion efficiency of the intraband transition.

It should be also mentioned that since a CW probe was used here to determine the reflection efficiency, only 0.1% of CW wave was converted. All power of the CW probe at times between the pump pulses adds to the average detected power, thus lowering the “apparent” conversion efficiency while the “effective” conversion efficiency of the probe wave packets participating in the conversion process remains at 35% for 6 ps pump pulses. For probe pulses or pulse trains that have durations equal to the interaction time with FC front we expect the conversion efficiency to approach 100%.”

Additionally we added Figure. S5 to the supplemental materials.

[2] In Fig. 2(b), the group index obtained in experiment is provided. Is it consistent with the theoretical calculation of $\frac{c}{L} \times \frac{d\psi}{d\omega}$?

Authors: The reviewer is right, the group velocity was measured by evaluating the frequency dependent phase variation at the output of the slow light waveguide. This information was obtained with an external Mach-Zehnder interferometer as in Ref. (A. Gomez-Iglesias, D. O'Brien, L. O'Faolain, A. Miller, and T. F. Krauss, "Direct measurement of the group index of photonic crystal waveguides via Fourier transform spectral interferometry," Appl. Phys. Lett. 90, 261107 (2007)). The following text was added to supplemental material, page 1:

"The group velocity was measured using an external Mach-Zehnder interferometer³³"

[3] (Line 349) The author should further demonstrate that why the front has a graded refractive index, linearly growing along the negative z-direction.

Authors: We have added a discussion on the front profile in the main text, page 14:

"The gradual profile of the front can be explained as follows. The FC generation rate via TPA is proportional to local pump power squared (see supplemental materials, section 2). The generation rate defines the slope of the FC concentration in time, and, taking the group velocity of the pump into account, in space. Thus the maximal slope corresponds to the position of the peak power of the pump and the front duration corresponds to the duration of the pump power squared distribution. Disregarding the FC decay, the FC concentration function is the integral of the pump power squared function. In this consideration we also neglect Kerr effect as for 6 ps pulses the FC dispersion is the dominant mechanism of the refractive index change³¹."

[4] The author should simply discuss potential application of optical switching.

Authors: We discuss the potential applications of FC front induced frequency shifting for wavelength division multiplexing (WDM) systems in the last paragraph of the main text before conclusions.

[5] The author should discuss influence of the decay of pump power on functions of the slow-light waveguide.

Authors: We thank the reviewer for his comment. We have mentioned before in the previous manuscript the influence of pump power decay in lines 403-410. We have added now pump decay into our simulations (see supplementary materials, section 5).

We have added a discussion on the pump power decay in the main text, page 17:

"We estimate total losses of ≈ 4 dB inside our slow light waveguide as a result of ≈ 2 dB linear losses³³, and ≈ 2 dB nonlinear losses due to FC absorption and TPA¹². As the FC generation rate via TPA is proportional to the local pump power squared, then the index front has 8 dB

exponential decay during propagation inside the waveguide. Such losses reduce the total effect of intraband transitions along the full waveguide length. The losses of the pump in combination with the incomplete transitions, lead to the spectral broadening of the blue shifted probe shown in Fig. 3”.

Furthermore, we have discussed this point in more details in section 2 of the supplemental materials.

[6] The author should clarify why the cross-talk problem can be overcome in the present design.

Authors: Our use of the term cross-talk was not precise enough. We meant the cross-talk between the high power pump and the signal. In case of FWM, due to the proximity to the signal frequency pump can have frequency components at the signal frequency. In case of FC front, the pump can be positioned far enough in frequency to avoid this cross-talk. We have revised the text on page 22 as following:

“In addition, for switching in WDM systems, the pump can be positioned outside of the signal frequency window, such that pump has no frequency components at the signal frequency. In comparison, the FWM approach requires the pump frequency always to lie between signal frequency and shifted frequency thus additional measures would be required to filter the pump out of the signal window and spectral cross-talk between the high power pump and the signal is difficult to avoid.”

Reviewer #2 (Remarks to the Author):

The authors report on frequency conversion at telecom wavelengths in slow-light photonic crystal waveguides, employing free carriers produced by two photon absorption of a ps pump pulse. The mechanism for frequency conversion is the interaction of a probe pulse with the plasma generated by the pump, which shifts the waveguide dispersion resulting in a photonic intraband transition. The authors quote a 35% forward reflection efficiency at a peak power of 6.2 W, equivalent to 1.2×10^9 W/cm² incident on the waveguide. The 35% efficiency is the ratio between the actual conversion efficiency, namely 3.5×10^{-4} , and the maximally transformed probe power fraction of 1×10^{-3} for the given experimental conditions. The authors claim the mechanism of intraband transitions to be a promising possibility for all-optical signal routing in WDM networks.

While this manuscript reports on advanced and well performed experiments, with interesting results for frequency conversion in slow-light PC waveguides, I do not believe this work to be important enough for publication in a Nature journal. My specific comments are objections are detailed below:

1. The title is “Reflection from a relativistic free carrier plasma mirror...”, the abstract and the

introduction start with a description of free carrier plasmas used as tunable mirrors at solid surfaces. Yet this work refers to a totally different situation, namely frequency conversion through interaction between a probe and a pulse beam, both propagating in a photonic crystal waveguide. It is clear that the threshold for frequency conversion is lowered by orders of magnitude compared to the surface geometry, as the beams are laterally confined and propagate for a nearly mm length across the waveguide. Moreover, “reflection” sounds as a misnomer in the present configuration, as the frequency-shifted pulse is propagating in the forward direction and is not “reflected”. I find the abstract and the introduction to be misleading in this respect.

Authors: We thank the reviewer for bringing our intention to this point. We changed the title, abstract and introduction of the manuscript. We omitted the discussion of relativistic plasma mirrors and focused on the discussion of ionization fronts and indirect photonic transitions.

We consider a system where the probe wave in the laboratory frame is approached by the co-propagating front and reflected from the front in the forward direction. To avoid confusion we now named the observed effect as “forward reflection”. However, it should be mentioned that in the reference frame moving with the front the probe is reflected and changes its direction of propagation.

2. The results for frequency conversion by intraband transitions in a slow-light PC waveguide are interesting, but not as groundbreaking as the introduction would suggest. Indeed, this work builds on other papers on the same topic, both by the same and other authors (e.g. Refs. 27, 28, 31). Specifically, the configuration for frequency conversion is similar to that of Ref. 27. The results for the conversion efficiency are indeed good compared to other systems, but in the end the absolute conversion efficiency is lower than 10^{-3} and most of the probe intensity is lost by absorption. It is difficult to envisage how this mechanism could be employed for all-optical routing.

Authors: We thank the reviewer for his comment. We stand by the claim that the intraband photonic transition and thus the forward reflection of the probe by a FC front is an effect that was not demonstrated so far. These results go well beyond the state of the art, including our prior work.

The CW probe was used here to determine the reflection efficiency. The fact that the “apparent” conversion efficiency is only 0.1% with respect to the incident CW power is not indicative of the “effective” conversion efficiency. All power of the CW probe at times between the pump pulses adds to the average detected power, thus lowering the “apparent” conversion efficiency while the “effective” conversion efficiency of the probe wave packets participating in the conversion process remains at 35% for 6 ps pump pulses. An identical approach was used in important event horizon experiments that we mention in the text (Philbin, T. G. *et al. Science* 319, 1367 (2008), Webb, K. E. *et al. Nature Communications* 5, 4969 (2014), Ciret, C. *et al. Opt. Express* 24, 114–124 (2016)). For probe pulses or pulse trains that have durations equal to the interaction time with FC front we expect the conversion efficiency to approach 100%. We have also added following text on page 21:

“In case of a sharper front, the time spent by the wave packets inside the front Δt will be shorter and the wave packets will experience less FC absorption. We show by simulation [see supplemental materials, Fig. S5], that using 1 ps pump pulse with the same peak power at the input of the slow light waveguide two times higher efficiency for the intraband transition could be obtained than in case of 6 ps pump pulse. The forward reflected probe does not penetrate the region with high FC concentration behind the pump pulse, thus FC absorption does not pose a fundamental limit on the conversion efficiency of the intraband transition.

It should be also mentioned that since a CW probe was used here to determine the reflection efficiency, only 0.1% of CW wave was converted. All power of the CW probe at times between the pump pulses adds to the average detected power, thus lowering the “apparent” conversion efficiency while the “effective” conversion efficiency of the probe wave packets participating in the conversion process remains at 35% for 6 ps pump pulses. For probe pulses or pulse trains that have durations equal to the interaction time with FC front we expect the conversion efficiency to approach 100%.”

3. I do not really understand why the authors present their results by writing that “we have experimentally demonstrated a 35% optical wave reflection and -3.4 nm wavelength shift by interaction with a relativistic FC plasma mirror...” The optical wave is not reflected, and the conversion efficiency is 3.4E-3.

Authors: We addressed this question in the comments to points 1 and 2.

Other remarks:

4. What happens in the results if the pump power is increased beyond 6.2 W? Can the conversion efficiency be further increased, or do absorption losses prevail?

Authors: If the pump power is increased beyond 6.2 W, the reflection coefficient will not continue to rise as the sufficient band shift for inducing the intraband transition is already obtained. However, the free carrier absorption could increase which in turn reduces the conversion efficiency.

Furthermore, the conversion efficiency can be increased if the time spent by the wave packets inside the front Δt is shorter and hence the wave packets will experience less free carrier absorption. This can be achieved by using a sharper front. We showed by simulation that using 1 ps pump pulse with the same peak power at the input of the waveguide will lead to a reflection efficiency of $\approx 70\%$. We added to the following text to the manuscript in page 21:

“In case of a sharper front, the time spent by the wave packets inside the front Δt will be shorter and the wave packets will experience less FC absorption. We show by simulation [see supplemental materials, Fig. S5], that using 1 ps pump pulse with the same peak power at the input of the slow light waveguide two times higher efficiency for the intraband transition could be obtained than in case of 6 ps pump pulse. The forward reflected probe does not

penetrate the region with high FC concentration behind the pump pulse, thus FC absorption does not pose a fundamental limit on the conversion efficiency of the intraband transition.

It should be also mentioned that since a CW probe was used here to determine the reflection efficiency, only 0.1% of CW wave was converted. All power of the CW probe at times between the pump pulses adds to the average detected power, thus lowering the “apparent” conversion efficiency while the “effective” conversion efficiency of the probe wave packets participating in the conversion process remains at 35% for 6 ps pump pulses. For probe pulses or pulse trains that have durations equal to the interaction time with FC front we expect the conversion efficiency to approach 100%.”.

Additionally we added Figure. S5 to the supplemental materials.

5. Figure 4 is very hard to read, black and blue colors with such thin lines are difficult to distinguish.

Authors: Perhaps the reviewer means Figure 3 as only their thin blue lines were used. We have changed the color scheme such that the probe without the pump and with the pump can be better distinguished.

Reviewer #3 (Remarks to the Author):

The paper by Gaafar et al. proposes an experimental demonstration on a photonic crystal waveguide platform of a pulse routing effect analogous to a reflection from a "free carrier plasma mirror". Its main claim, notably with respect to the previous work by Baba's team (Kondo 2014 PRL) on Dynamic Wavelength Conversion, is to achieve intraband transitions rather than interband ones, owing to the large band shift and the specific elbow-shaped bands. The free-carrier origin is, in my understanding, the reason why the term "reflection" is used, even though the pulse is forward kicked rather than backward kicked by the moving carrier plasma.

The paper is not fully convincing, and has a few weaknesses however, some more severe than the others, with respect to its claims. In my opinion, it cannot be published as such unless the main weaknesses are properly and sufficiently addressed.

From the point of view of wording first, the paper title and some introductory paragraphs boast the "relativistic" aspect. It certainly adds appeal to the narrative and admittedly the situation bears a fair relationship, but it should be much weighted with the peculiarities of this experiment : the velocity is only $v_g \sim c/40$, much smaller than c . Furthermore, a moving front of nonlinear matter modification due to somehow relaxed electrons in the carrier plasma has been commonplace in nonlinear optics, be it in fiber, in crystals, or in nanophotonic devices. This contrasts, for instance, with what the recent Ref. 7 states for

electrons of relativistic mirrors: "When pulled outwards, they form relativistic electron jets (red arrow in Fig. 1a) that are responsible for the ROM[Relativistic Oscillating Mirror] attosecond pulse emission." It is not clear whether that part of the relativistic physics is present in the reported work. So, much more care would be needed and possibly the title could be modified if the case is not made more strongly towards the analogy. Ref.28 by Kondo and Baba does not invoke relativistic effects at all, while the mechanisms could be with little reduction claimed to be "half the same" (interband vs. intraband, say) insofar as plasma effects are concerned.

The fact that the experiment works in transmission rather than in reflection also causes some difficulty in pursuing the analogy. In balancing the discussion more clearly towards the "ionization fronts", the paper could gain more conviction than within the present unbalanced choice of emphasizing relativistic plasma mirrors.

Authors: We thank the reviewer for bringing our intention to this point. In the revised manuscript, we changed the title, abstract and introduction of the manuscript. We omitted the discussion of relativistic plasma mirrors and focused our discussion on ionization fronts and intraband transitions. The new title became "Reflection from a free carrier front via an intraband indirect photonic transition". We have also followed the reviewer's suggestion to use the term "forward reflection" to describe the presented effect.

However, the two main weaknesses of the paper are in my opinion the following ones :

- First, the paper (including the SI) is based on a single sample and a single length. Practitioners often make several samples side by side, it is hard to argue that the task of multiple sample fabrication was an obstacle. Given the limits on spectral behaviour entailed by the particular length choice (and this is according to the authors themselves: conversion is partial due to finite length etc., the role of dispersion should be interesting with different dispersions, line 466, etc.), the limit of a single sample experiment leaves more room for some alternative explanations (e.g. based on trapped/untrapped photons, possible role of irregularities on the waveguide causing some unsteady ionization front dynamics, etc.) than if an experiment with a different sample could take place. To some extent, changing the wavelength gives extra room, but the wavelength ranges where the expected phenomena unfold are very limited given the specific mechanism targeted and the associated dispersion demands. If a longer device of identical dispersion, for instance, would raise difficulties such as too low transmission, then there are some limits to the proposed scheme with state-of-the-art PhC waveguides, and they should be clearly and quantitatively acknowledged if not emphasized.

Authors: From the outset, we have produced waveguides of equal length. The production of new samples requires a very big effort as it includes many processing steps apart from e-beam lithography and etching. It includes deposition on polymer waveguides for inverted tapers and underetching of the slow light waveguide area. Although we are considering this step for a future phase of our research, we are currently limited to the 400 μ m length. Also based on our

experience of fabricated devices in other projects, the repeatability is high. The performance of this device lies within the normal ranges for this class of device.

We have measured the group velocity of the sample and have reconstructed the band diagram of the obtained slow light waveguide. Using this band diagram and the expected refractive index shift generated by the pump pulse we have simulated the trajectories of the probe wave packets. The results are included in the main text and supplementary materials, now. The simulation results are obtained without fitting parameters and reproduce the frequency distribution and conversion efficiency obtained in the experiment. We hope that the comparison to simulations will increase confidence in our results and would allow the reviewer to support our line of argument.

We have added a discussion on the pump power decay in the main text, page 17:

“We estimate total losses of ≈ 4 dB inside our slow light waveguide as a result of ≈ 2 dB linear losses³³, and ≈ 2 dB nonlinear losses due to FC absorption and TPA¹². As the FC generation rate via TPA is proportional to the local pump power squared, then the index front has 8dB exponential decay during propagation inside the waveguide. Such losses reduce the total effect of intraband transitions along the full waveguide length. The losses of the pump in combination with the incomplete transitions, lead to the spectral broadening of the blue shifted probe shown in Fig. 3”.

Furthermore, we have discussed this point in more details in section 2 of the supplemental materials.

- Secondly, the methodology is rather weak when it comes to analysing the resulting probe spectra, mainly those of Fig.3. Experimentally, was there no way to on/off modulate the probe beam (at 100 MHz) to avoid, say, 90% of the background? More importantly, an attempt to analyse the spectral shape should be explicitly made and rendered as e.g. graphs/bars with one point associated to each spectrum/pump power, so that other investigators in the future can check what is going on based on the sole non quantitative spectral shape appearance. Indicating the pulse "centers" (vaguely defined) by arrows in such a smooth spectral context is good at a preliminary stage (for a post-deadline paper, for instance) but more explicit analysis is commendable for a high-standard paper.

Authors: We thank the reviewer and agree with his comment. It was our decision to use CW wave for demonstration purposes as it allows clearly defining the energy which is interacting with the front. An identical approach was used in important event horizon experiments that we mention in the text (Philbin, T. G. *et al. Science* 319, 1367 (2008), Webb, K. E. *et al. Nature Communications* 5, 4969 (2014), Ciret, C. *et al. Opt. Express* 24, 114–124 (2016)).

It should be mentioned that since a CW probe was used here to determine the reflection efficiency, only 0.1% of CW wave was converted. All power of the CW probe at times between the pump pulses adds to the average detected power, thus lowering the “apparent” conversion efficiency while the “effective” conversion efficiency of the probe wave packets

participating in the conversion process remains at 35% for 6 ps pump pulses. For probe pulses or pulse trains that have durations equal to the interaction time with FC front we expect the conversion efficiency to approach 100%.

If we would modulate the CW wave, the transmitted average signal power will reduce, but the converted signal will have the same average power as the same portion of the CW light will be interacting with the front. The signal to noise ratio will not improve in this case.

The converted signal would be much higher above the noise floor if we would use a pulsed probe. But in this case the calculation of the conversion efficiency would be more difficult, as the delay between pump and probe and its jitter would be additional parameters. Even our CW case is complex enough as our schematic Fig. 3 shows. The different wave packets can accumulate incomplete transitions resulting in a complex frequency distribution of the converted signal. We can account for that when we use CW wave as all packets have the same energy. In case of the pulsed probe the complexity of the frequency response will be difficult to interpret.

To analyse the data we have added results from ray tracing simulations to the main text (page 20-21) and supplemental material (section 5). The simulation results obtained without fitting parameters reproduce the frequency distribution and conversion efficiency obtained in the experiment. Thus it confirms our claim of 35% reflection at the free carrier front. Further improvements can be achieved with shorter pump pulses as confirmed by simulations.

- A few other suggestions to attain an easier reading are the following ones:

1) it would be good to still attempt (in the SI for instance) to represent the band structure at rest and with carrier plasma, even if it is graphically awkward (local zooms at elbows?). The present effort to grasp the diagrams with renormalized slope are truly difficult even for a scientist familiar with PhC band structures.

Authors: We added the band structure with and without refractive index change as a new figure in the supplemental materials [Fig. S3].

2) Δ_n is negative, so Fig.4(b) is also awkward to read as the usual trend in optics is that $\Delta_n > 0$ slows down light, while here $\Delta_n < 0$ speeds it up. Check and possibly amend the "index axis" of Fig.4b to clarify the sign and role of Δ_n .

Authors: Thank for noticing this error. We have corrected Fig. 4(b) right.

3) Fig.4c, the solid blue dot is still interband, the caption is probably misleading (or the color ...).

Authors: Correct, we have changed it to dashed dot.

4) Typos : line 332 "we chose to view *THE* cw probe".

Some other choices are idiosyncratic (to my eyes) and not the best:

--The use of the verb "employ" to describe the plasma rising front "finite steepness".

--The use of "a.k.a." on line 358.

--The sentence lines 385-386 : "Then, ... such is ..." looks awkward.

--Lines 463 464 : "thus caus*ING* a bigger cross-talk problem. [And] *The* third advantage ... " (suppress "And").

Authors: We thank the reviewer for this comments, we took them into account in the revised version.

Overall, my recommendation, again, is not to publish as such, essentially because the paper too often indulges in easier lines of analysis for critical aspects: the relevance of the relativistic mirror comparison, the degree of conviction of the conclusion drawn on a single sample, and the rather qualitative analysis of the spectra (even if no new science is involved in fitting waveforms among themselves or against some specific profiles).

But it must be recognized that a distinctive intraband transition was indeed observed which makes a possibly large difference with respect to earlier work, so that a revision with substantially enhanced analysis with a more appropriate relationship to current trendy topics in optics could be envisioned.

Reviewers' Comments:

Reviewer #1:

Remarks to the Author:

Authors replied the reviewers' comments well. This manuscript can be accepted.

Reviewer #2:

Remarks to the Author:

The manuscript has been thoroughly revised by the authors, my main comments have been met by several changes in the title, abstract, and in the text. I do agree that the demonstration of an intraband photonic transition is an important new result. I can recommend this work for publication.

Reviewer #3:

Remarks to the Author:

The paper revision now retracts clearly from the claim of "ultrarelativistic mirrors" etc. It is indeed a wise move. After all, the report is concerned with nothing but a pump-probe experiment with pulses (/virtual wave packets) co-propagating at close (and interacting) velocities, a configuration that, per se, has been around in photonics since a long time, even if I am not aware of a particular string of experiments reflecting a similar photonic transition in a slow-group velocity context. So originality can be granted.

The main revision that has to do with methodology (previously deemed poor) and clarification of the mechanisms and the margins is the couple of modelling exercises substantiated by Fig. S4 and Fig.S5 of the SI.

A successful part of the effort is that it is shown that it is possible to visualize the modest velocity deviations even without taking out a bias slope ($x=vt$), i.e. even without reasoning in a moving frame (but in S.I. with plenty of room for explanations, this would have been an at least equally valuable path in my view).

However, the exercise seems stuck midway between the start of the intellectual effort and the full-fledged effort needed for the targeted audience. Given the quality and number of the authors, a large doubt is introduced as regards the existence of a solid understanding, as detailed below. So it seems that in a first step, the authors rushed to make an appealing claim ("ultrarelativistic plasma mirrors") on a rather thin basis, but now they are retreating without being able to exhibit a robust basis beneath the appealing story.

In detail, what was found disturbing in the simulation are the following points (from benign to serious):

- No effort to visualize the many elements needed to grasp the picture : no visualization of the front by shades as was done in the main text, so more efforts are needed to read the figures and understand why grey lines (wave packets out of the front) are where they are. It also demands some effort to check how much at 1531.5 nm the front and pulse velocities coincide (since outer wave packets never have a chance to enter the front: grey lines remain grey), whereas a precise enough reminder of group index dispersion would be welcome. Similarly, a reminder that the 40 ps simulation time is associated with the guide physical length of 400 microns (albeit at the pump velocity, it makes for up to a ~20% longer path at the shifter probe velocity). These are the main limits that frame the experiment and its representation and they are not visualized.

- No effort to clearly account for $\Delta\omega$, which is left in reduced units (0.00007). I could not find an easy (nor uneasy) correspondance with the actual values (fractions of 200 THz...). This is annoying.

- No explanation of the grossly discretized output spectrum. Why is a "binning" by coarse 1 nm intervals performed, much more than experimental resolution (say 0.2-0.4 nm depending on the noise limit taken into account to safely identify a signal)? The reader is left with assumptions related to the 0.9 ps interval of successive wave packets chosen as illustration in this geometric optics model. But if the model only has to iterate over 400 steps of 0.1 ps each, there is no apparent difficulty in tracing the fate of five hundred rays and make some decent histogram with fine meshing adapted to the experiment.

So, the lack of a convincing appearance of the simulation, even though their basis seems correct, introduces too much doubts and lessens the degree of conviction that the authors have genuinely grasped what their experiment is doing quantitatively. Or for some reason, they do not want to spoil material for further publication elsewhere, etc. Whatever the exact rationale for the appearance of the results, they entail that on a further round of revision, the authors would have to retreat once more to a less convincing line of arguments, as weaknesses could be hidden by the above points (poor $\Delta\omega$ calibration, difficulty to refine the spectral steps in modelling). The review process is not intended to prolong the effort till the authors and reviewers reach a common point in a "contra-propagative" way, the former retreating and the latter bringing incentives to progress. In the reverse case, co-construction of progresses with a fertilizing role of the reviewer(s), I would be much more happy to prolong the review by an extra round.

However, in the present case, my conclusion from the awkward output of this round is that while there is a large potential for the authors, it cannot be reached within the ongoing review process.

Therefore I do not recommend the paper for publication.

Reviewer #3 (Remarks to the Author):

The paper revision now retracts clearly from the claim of "ultrarelativistic mirrors" etc. It is indeed a wise move. After all, the report is concerned with nothing but a pump-probe experiment with pulses (/virtual wave packets) co-propagating at close (and interacting) velocities, a configuration that, per se, has been around in photonics since a long time, even if I am not aware of a particular string of experiments reflecting a similar photonic transition in a slow-group velocity context. So originality can be granted.

The main revision that has to do with methodology (previously deemed poor) and clarification of the mechanisms and the margins is the couple of modelling exercises substantiated by Fig. S4 and Fig.S5 of the SI.

A successful part of the effort is that it is shown that it is possible to visualize the modest velocity deviations even without taking out a bias slope ($x=vt$), i.e. even without reasoning in a moving frame (but in S.I. with plenty of room for explanations, this would have been an at least equally valuable path in my view).

However, the exercise seems stuck midway between the start of the intellectual effort and the full-fledged effort needed for the targeted audience. Given the quality and number of the authors, a large doubt is introduced as regards the existence of a solid understanding, as detailed below. So it seems that in a first step, the authors rushed to make a appealing claim ("ultrarelativistic plasma mirrors") on a rather thin basis, but now they are retreating without being able to exhibit a robust basis beneath the appealing story.

In detail, what was found disturbing in the simulation are the following points (from benign to serious):

- No effort to visualize the many elements needed to grasp the picture : no visualization of the front by shades as was done in the main text, so more efforts are needed to read the figures and understand why grey lines (wave packets out of the front) are where they are. It also demands some effort to check how much at 1531.5 nm the front and pulse velocities coincide (since outer wave packets never have a chance to enter the front: grey lines remain grey), whereas a precise enough reminder of group index dispersion would be welcome. Similarly, a reminder that the 40 ps simulation time is associated with the guide physical length of 400 microns (albeit at the pump velocity, it makes for up to a ~20% longer path at the shifter probe velocity). These are the main limits that frame the experiment and its representation and they are not visualized.

Authors: We thank the reviewer for his comments. We took them into account in the revised version.

1. Even though the reviewer mentioned that the packet trajectories can be seen without taking out a bias slope ($x= vt$), we still think the close proximity of the group velocities of the pump and the probe might lead to confusion in the interpretation. Thus, in order to see the velocity deviations of the wave packets before and after interaction with the front more clearly, we subtracted now the slope $x= vt$ in Fig. 5(b) of the main text. Nonetheless, as suggested, we present the original curves in the supplementary material [Figs. S4 and S5].

We have added to the following text to the main manuscript in pages 20 and 21:

“The trajectories of the probe wave packets with respect to the front velocity, which are represented by black lines, before and after the interaction with the index front

inside the 400 μm long waveguide, are shown in Fig. 5(b). Due to the small group velocity mismatch between the initial probe wave packets and the index front, the trajectories are difficult to be shown clearly in their original form. Thus, the slope corrected curves are presented with time t' calculated as $t' = t - x/v_f$. In this case, the two orange lines represent the index front with a rise time of 6 ps, appearing static in this representation, whereas the shading represents the front refractive index changes exponentially decaying during propagation inside the waveguide. For comparison, original uncorrected curves are presented in the supplementary material [Figs. S4 and S5].

Here we show the trajectories of only four wave packets for clarity. Solid black lines represent those wave packets that undergo complete transitions, while the dashed lines depict the wave packets which only undergo incomplete transitions. We can see clearly, that after interaction with the front some wave packets are accelerated by the front and bounced in the forward direction (solid lines) and some other wave packets still have velocities smaller than the front (dashed lines). Additionally, we can observe that some wave packets starting initially inside the front are accelerated but do not have the enough time to leave the front.”

2. The grey lines in the previous version represented those wave packets which do not interact with the front, either they left the waveguide before the front reach them, or they came after the front and saw the already switched structure. To avoid confusion, we removed these wave packets from the current version, as they do not contribute to any frequency shift, and showed only those wave packets which undergo incomplete and complete transitions. We limited the figure to only four representative trajectories, not to overload the figure. The new versions are now presented as Fig. 5 (b) in the

main manuscript, as well as Figs. S4(b), S4(d) and S5(b) in the supplemental materials.

3. The shading for the index front is now included in Fig. 5(b) in the main manuscript, Figs. S4(b), S4(d) and S5(b) in the supplemental materials which is decaying with propagation in the waveguide.
4. The propagation length and time are additionally discussed in connection with the new Figs. S4 and S5 in the supplemental materials. We added additionally the following text in page 8 of the supplemental materials:

“Black lines represent those wave packets that interacting with the front and undergo interband and intraband transitions. The two magenta lines illustrate the extent of the index front with a slope equal to the group velocity of the pump and with a rising time of 6 ps. The 40 ps simulation time is associated with time spent by the front with a group index of 30 inside the 400 μm long waveguide.”

And the following sentence in page 9 in the supplemental materials:

“(the same as Fig. 5 in the main text but without correcting the time by the slope of the pump group velocity)”.

- No effort to clearly account for $\Delta\omega$, which is left in reduced units (0.00007). I could not find an easy (nor uneasy) correspondance with the actual values (fractions of 200 THz...). This is annoying.

Authors: We thank the reviewer and agree with his comment. We have represented the $\Delta\omega$ now in THz units in the revised version. We also added the following text in page 20 of the main manuscript:

“Figure 5 presents the simulation results for $\Delta\omega_{shift}$ of 0.12 THz (0.9 nm) at the input corresponding to refractive index change of $\approx -3 \cdot 10^{-3}$.”

- No explanation of the grossly discretized output spectrum. Why is a "binning" by coarse 1 nm intervals performed, much more than experimental resolution (say 0.2-0.4 nm depending on the noise limit taken into account to safely identify a signal)?

Authors: The ray optics approximation presented here does not take into account the spectral width of the wave packet and interference between packets. It considers the wave packet as a point in time, space and frequency. The simulation identifies the final frequency of the wave packet after interaction with the front, which corresponds to the central frequency of a wave packet in wave optics treatment. To compare this simulation to the experiment, we need to calculate how many wave packets are obtained in the certain frequency interval. Previously we used histogram type of curve with 1 nm resolution. We agree that this leads to a very coarse representation. We decided now to use a 1 nm frequency interval and scanned the center frequency with a finer step, the same as was used in the experiment with an optical spectrum analyzer.

These reasons for using 1 nm interval are now additionally presented in the paragraph where Fig. 5(a) is introduced. The following text is added in page 20 of the main manuscript:

“Figure 5(a) shows the calculated average power of the wave packets in frequency intervals of 1 nm scanned over the spectrum, the same resolution that was used in the experiment with an optical spectrum analyzer.”

The reader is left with assumptions related to the 0.9 ps interval of successive wave packets chosen as illustration in this geometric optics model. But if the model only has to iterate over

400 steps of 0.1 ps each, there is no apparent difficulty in tracing the fate of five hundred rays and make some decent histogram with fine meshing adapted to the experiment.

Authors: In the histogram, the number of simulated wave packets was already 120 as we used 0.1 ps spacing for the calculation and 0.9 ps spacing for the graphical representation. To avoid confusing histogram representation we have now scanned the spectrum with 1 nm bandwidth, similar to the experiment. We now obtain a smooth simulation spectrum with a resolution of 1 nm.

We have added the following sentence in page 21 of the main manuscript:

“For the spectral evaluation we have launched 120 equally spaced wave packets with starting time in the interval $t' = [-5 \text{ ps}, 7 \text{ ps}]$ ”.

Furthermore, we added the following text in page 20 of the main manuscript:

“Figure 5(a) shows the calculated average power of the wave packets in frequency intervals of 1 nm scanned over the spectrum, the same resolution that was used in the experiment with an optical spectrum analyzer.”

So, the lack of a convincing appearance of the simulation, even though their basis seems correct, introduces too much doubts and lessens the degree of conviction that the authors have genuinely grasped what their experiment is doing quantitatively. Or for some reason, they do not want to spoil material for further publication elsewhere, etc.

Authors: The simulation results we now present are clearly limited by the ray optic approximation and therefore, an exact spectral correspondence cannot be expected. At the same time, we have obtained a good correspondence between simulation and experiment in terms of spectral power density.

Whatever the exact rationale for the appearance of the results, they entail that on a further round of revision, the authors would have to retreat once more to a less convincing line of arguments, as weaknesses could be hidden by the above points (poor δ/ω calibration, difficulty to refine the spectral steps in modelling). The review process is not intended to prolong the effort till the authors and reviewers reach a common point in a "contra-propagative" way, the former retreating and the latter bringing incentives to progress. In the reverse case, co-construction of progresses with a fertilizing role of the reviewer(s), I would be much more happy to prolong the review by an extra round.

However, in the present case, my conclusion from the awkward output of this round is that while there is a large potential for the authors, it cannot be reached within the ongoing review process.

Therefore I do not recommend the paper for publication.

REVIEWERS' COMMENTS:

Reviewer #3 (Remarks to the Author):

Even though the style of the answer remains perfectly neutral, I found that the new content was clearly moving the paper forward from the previous version by more than an incremental amount.

Having a continuous version of the spectral modelling (albeit with simplifications in its derivation) and providing a clear view of the frequency shifts (fractions of THz) behind it offered more than fresh air : a genuine satisfaction that the authors themselves were better able to explain how phenomena unfold in their apparently simple system, qualitatively beyond the former step.

In particular, the new emphasis on the decay of the strong pulse upon propagation is a key element for future research in the area, one that was nearly absent (or well hidden) in the general frame of the previous version (even though it was duly mentioned in the report). It is well rendered by the red-to-yellow grading in Fig.5. (It would be possible, in follow-up articles, to also include it in the equivalent of Fig.4. All in all, this limitation is reminiscent, broadly speaking, of a gain-bandwidth product ultimately limiting what the best device design can offer once losses are accounted for).

The SI also offers a more comprehensive treatment, contrasting cases of insufficient and sufficient frequency shift for inducing transitions of interband nature.

While my former impression was obviously negative due to the failure of producing such data treatment and convincing modelling that were strongly needed in my view, I must now admit that the authors have achieved the step forward that I thought they were either reluctant or unable to achieve within a reasonable set of referral rounds.

I can now lift my strong objections and admit that the paper can be recommended for publication. In this new version, not only will it be much less misleading to the audience, but it will also be better suited to prompt new clever developments on a safer basis.

REVIEWERS' COMMENTS:

Reviewer #3 (Remarks to the Author):

Even though the style of the answer remains perfectly neutral, I found that the new content was clearly moving the paper forward from the previous version by more than an incremental amount.

Having a continuous version of the spectral modelling (albeit with simplifications in its derivation) and providing a clear view of the frequency shifts (fractions of THz) behind it offered more than fresh air : a genuine satisfaction that the authors themselves were better able to explain how phenomena unfold in their apparently simple system, qualitatively beyond the former step.

In particular, the new emphasis on the decay of the strong pulse upon propagation is a key element for future research in the area, one that was nearly absent (or well hidden) in the general frame of the previous version (even though it was duly mentioned in the report). It is well rendered by the red-to-yellow grading in Fig.5. (It would be possible, in follow-up articles, to also include it in the equivalent of Fig.4. All in all, this limitation is reminiscent, broadly speaking, of a gain-bandwidth product ultimately limiting what the best device design can offer once losses are accounted for).

The SI also offers a more comprehensive treatment, contrasting cases of insufficient and sufficient frequency shift for inducing transitions of interband nature.

While my former impression was obviously negative due to the failure of producing such data treatment and convincing modelling that were strongly needed in my view, I must now admit that the authors have achieved the step forward that I thought they were either reluctant or unable to achieve within a reasonable set of referral rounds.

I can now lift my strong objections and admit that the paper can be recommended for publication. In this new version, not only will it be much less misleading to the audience, but it will also be better suited to prompt new clever developments on a safer basis.

Authors: We thank reviewer for recommending our work for publication in Nature Communication.